computational chemistry/physical chemistry/
spectroscopy

nonlinear optical chemistry, π-conjugated linkers,
density functional theory

**Authors for correspondence:**
Muhammad Khalid
e-mail: khalid@iq.usp.br
Muhammad Safwan Akram
e-mail: safwan.akram@tees.ac.uk

This article has been edited by the Royal Society of Chemistry, including the commissioning, peer review process and editorial aspects up to the point of acceptance.

# Structural modulation of π-conjugated linkers in D–π–A dyes based on triphenylamine dicyanovinylene framework to explore the NLO properties

Muhammad Khalid[1], Muhammad Usman Khan[2],
Iqra Shafiq[1], Riaz Hussain[2], Akbar Ali[3],
Muhammad Imran[4], Ataualpa A. C. Braga[5],
Muhammad Fayyaz ur Rehman[3] and Muhammad
Safwan Akram[6,7]

[1]Department of Chemistry, Khawaja Fareed University of Engineering and Information Technology, Rahim Yar Khan 64200, Pakistan
[2]Department of Chemistry, University of Okara, Okara 56300, Pakistan
[3]Institute of Chemistry, University of Sargodha, Sargodha 40100, Pakistan
[4]Department of Chemistry, Faculty of Science, King Khalid University, PO Box 9004, Abha 61413, Saudi Arabia
[5]Departamento de Química Fundamental, Instituto de Química, Universidade de São Paulo, Avenida Prof. LineuPrestes, 748, São Paulo 05508-000, Brazil
[6]School of Health and Life Sciences, Teesside University, Middlesbrough TS1 3BA, UK
[7]National Horizons Centre, Teesside University, Darlington DL1 1HG, UK

MK, 0000-0002-1899-5689; MFuR, 0000-0003-2901-7212;
MSA, 0000-0001-9706-3152

A donor–π–acceptor type series of Triphenylamine–dicyanovinylene-based chromophores (**DPMN1–DPMN11**) was designed theoretically by the structural tailoring of π-linkers of experimentally synthesized molecules **DTTh** and **DTTz** to exploit changes in the optical properties and their nonlinear optical materials (NLO) behaviour. Density functional theory (DFT) computations were employed to understand the electronic structures, absorption spectra, charge transfer phenomena and the influence of these structural modifications on NLO properties. Interestingly, all investigated chromophores exhibited lower band gap (2.22–2.60 eV) with broad absorption spectra in the visible region, reflecting the remarkable NLO response. Furthermore, natural bond orbital (NBO) findings revealed a strong push–pull mechanism in **DPMN1–DPMN11** as donor and

π-conjugates exhibited positive, while all acceptors showed negative values. Examination of electronic transitions from donor to acceptor moieties via π-conjugated linkers revealed greater linear ($\langle\alpha\rangle$ = 526.536–641.756 a.u.) and nonlinear ($\beta_{tot}$ = 51 313.8–314 412.661 a.u.) response. It was noted that the chromophores containing imidazole in the second p-linker expressed greater hyperpolarizability when compared with the ones containing pyrrole. This study reveals that by controlling the type of π-spacers, interesting metal-free NLO materials can be designed, which can be valuable for the hi-tech NLO applications.

# 1. Introduction

Nonlinear optical materials (NLO) have a promising role in electro-optics for signal-analysis, fibreoptics, telecommunication and information technology [1–3]. NLO compounds derived from the organic framework have become the subject of contemporaneous investigations [4–6], because of their effortless reaction chemistry, low cost development and ability to tolerate structural modifications to allow for a unique NLO response. These NLO attributes of materials are underpinned by transfer of intramolecular charge (ICT) mainly generated from donor to acceptor units via π-conjugated linkers [7,8]. Experimental and computational data show that the broad second-order NLO response can arise from assembling robust donor (D) and acceptor (A) groups positioned at opposite sides of a π-linker, i.e. D–A, D–π–A, D–π–D–π–A and A–π–D–π–A. Compounds, with delocalized π-electrons with D–π–A configuration, show enhanced charge transfer transitions [9,10].

In the last few years, a significant number of metal-free organic donor–acceptor complexes have been identified as NLO compounds with π-conjugated linkers that offer a path for better charge transfer of electrons in the presence of an electric field [11–16]. The literature is replete with unique chemical structures using charge transfer (CT) between the electron donor and the withdrawal group to build new donor–π–acceptor systems that can minimize the bandgap and regulate the transitions using various donor or acceptor moieties with large first hyperpolarizability values ($\beta_{total}$) [17]. However, limitations exist in the development of certain essential features for wider acceptability and successful industrial applications. These include reliable synthesis, cost of manufacturing, ease of substitutions, tuneable absorption wavelength and the need for rapid skeleton modification [18,19]. We have previously published improved π bridges through double heteroaromatic rings and demonstrated their use in Triphenylamine–Dicyanovinylene dyes [17]. This manuscript demonstrates that π bridge modification is a convenient strategy to augment NLO response and to design novel NLO materials. The use of charge transfer (CT), between the electron donor and the withdrawal group, to build a new donor–π–acceptor system is capable of minimizing the bandgap and regulate the transitions using various donor or acceptor moieties with large first hyperpolarizability values ($\beta_{total}$) [11,13–15,20–22]. Inspired by these reported strategies, the electronic properties of D–π–A system with the new pi-conjugated system is introduced in the system consisting of 1H,1'H-2,2'-bipyrrole, 1,4 dihydropyrrolo[3,2-b]pyrrole, indole, 1,4-dihydroimidazo[4,5-d]imidazole, 3H,3'H-4,4'-biimidazole and benzimidazole referred to as initial π-spacer, and two conjugates, pyrrole and imidazole as second π-linker between the donor, Triphenylamine (TPA) and acceptor, Dicyanovinylene (DCV). TPA, a donor unit due to its capacity for electron donation and charge transfer, is used in many hole transport materials [23,24]. Eleven new D–π–A type, TPA-DCV dyes **DPMN1–DPMN11**, have been developed with various configurations of first and second π-conjugates. This empirical evaluation is appropriate for the estimation of the NLO properties and also for the investigation of the impact on the NLO activity of various π-conjugated linkers. Density functional theory (DFT) calculations for electronic characteristics, absorption spectrum, polarizability and first hyperpolarizability values were performed to compute the newly designed dyes (**DPMN1–DPMN11**). Hopefully, this research will act as a source to create new metal-free organic dyes with excellent NLO properties.

## 1.1. Computational procedure

DFT study was used to execute the electronic properties, charge transfer phenomena and nonlinear optical (NLO) behaviour of newly designed D–π–A systems (**DPMN1–DPMN11**). All calculations were carried out using the Gaussian 09 program [25] at the B3LYP/6–311+g (d, p) functional. Geometric optimization in entitled dyes (**DPMN1–DPMN11**) was carried out at the B3LYP/6–311+g (d, p) level. Frontier molecular orbitals (FMOs), global reactivity parameters (GRP), NLO and NBO (natural bond

orbital) analysis were performed at the same level. Absorption spectra of these organic dyes were computed via time-dependent density functional theory (TD-DFT) at Coulomb-attenuated hybrid exchange-correlation (CAM-B3LYP) level with the aforesaid basis set. Equations (1.1) and (1.2) can be used for calculating the polarizability $\langle \alpha \rangle$ and hyperpolarizability tensors ($\beta_{tot}$) of the entitled dyes [26]

$$\langle \alpha \rangle = 1/3(\alpha_{xx} + \alpha_{yy} + \alpha_{zz}) \tag{1.1}$$

and

$$\beta_{tot} = [(\beta_{xxx} + \beta_{xyy} + \beta_{xzz})^2 + (\beta_{yyy} + \beta_{yzz} + \beta_{yxx})^2 + (\beta_{zzz} + \beta_{zxx} + \beta_{xyz})^2]^{1/2} \tag{1.2}$$

A total of 10 hyperpolarizability tensors, $\beta_{xxx}$, $\beta_{xyy}$, $\beta_{xzz}$, $\beta_{yyy}$, $\beta_{xxy}$, $\beta_{yzz}$, $\beta_{zzz}$, $\beta_{xxz}$, $\beta_{yyz}$ and $\beta_{xyz}$, were achieved as an output from Gaussian file in the $x$, $y$ and $z$ directions [26]. Furthermore, Gauss View 5.0 [27], Avogadro [28] and Chemcraft [29,30] were used for the interpretation of output results.

# 2. Results and discussion

## 2.1. Structural modelling of D–π–A moieties

Screening of π-spacers plays a crucial role in donor–pi–acceptor type chromophores for the achievement of promising NLO response. The purpose of the current work is to design a novel Triphenylamine-dicyanovinylene-based promising NLO material by structural tailoring with various π-bridges and predict their photo physical, electronic and NLO behaviour for the latest optoelectronic applications. Herein, for the theoretical designing, a synthesized metal-free organic dye **D1** [31] is used. Our designed chromophores (**DPMN1–DPMN11**) are composed of three main parts: (i) Triphenylamine (TPA) as donor moiety, (ii) first and second pi-spacer that collectively played the role of bridge, and (iii) dicyanovinylene (DCV) acts as acceptor unit. A total of 11 molecules are designed using six π-conjugates: (i)(1H,1'H-2,2'-bipyrrole, (ii)1, 4dihydropyrrolo[3,2-b]pyrrole, (iii) indole, (iv)1,4-dihydroimidazo[4,5-d]imidazole, (v) 3H,3'H-4,4'-biimidazole, (vi) Benzimidazole) as the primary π-linker and two π-spacers (pyrrole and imidazole) as the second π-linkers as shown in figure 1. The dihedral angle between C–C–C in the benzene ring of TPA molecule is found to be 117° in all investigated molecules (**DPMN1–DPMN11**). The bond angle between C–C–N of 1,4-dihydropyrrolo[3,2-b]pyrrole attached to pyrole in **DPMN1** and imidazole in **DPMN2** is found to be 106° and 110°, respectively. In **DPMN3** and **DPMN4,** the C–N–N bond angle of 1,4-dihydroimidazo[4,5-d]imidazole attached to the TPA side beneze ring is found to be similar at 112°. The dihedral angles between C–C–N (pyrole) of **DPMN3** and in C–N–N (imidazole) of **DPMN4** are noted as 107° and 111°, respectively. In **DPMN5,** the dihedral bond angle between C–C–N of 1H,1'H-2,2'-bipyrrole is observed as 106°. A slight increase in bond angle to 110° in C–N–N of imidazole is noted when 1H,1'H-2,2'-bipyrrole is attached to imidazole unit. 104° dihedral bond angle is marked in C–N–N of 3H,3'H-4,4'-biimidazole in **DPMN6**. The dihedral angle of 107° is observed for C–C–N of pyrol unit in **DPMN6**. In compound **DPMN7**, 104° and 111° dihedral angles are found between C–N–N of 3H,3'H-4,4'-biimidazole and imidazole, respectively. In **DPMN8** and **DPMN9**, five-membered C–C–N of indole towards the TPA side exhibited a 108° dihedral angle, while benzene ring C–C–C of indole towards pyrol and imidazole exhibited a 120° dihedral angle. 106° and 110° dihedral angles are found between C–C–N in **DPMN10** pyrole unit and C–N–N in **DPMN11** imidazole unit, respectively.

Electronic transitions, ($\langle \alpha \rangle$ and $\beta_{tot}$), NBO analysis, spectral absorption analysis and light-harvesting efficiency (LHE) are performed by evaluating DFT and TD-DFT calculations for the exploration of NLO properties.

## 2.2. Electronic structure

FMO investigation is an excellent strategy for examining chemical stability and optoelectronic properties in investigated molecules [32]. The FMOs, i.e. HOMO (highest occupied molecular orbital) and LUMO (lowest unoccupied molecular orbital), play a significant part in absorption spectra and mechanical modelling of compounds [33]. For evaluating the strength, dynamic stability, softness, hardness and chemical reactivity of the designed compounds, the band gap ($E_{LUMO}$–$E_{HOMO}$) is the most significant factor [34]. The greater HOMO–LUMO distance within a molecule is associated with less reactivity, greater stability and hard molecule, while those molecules with small $E_{LUMO}$–$E_{HOMO}$ energy gap are

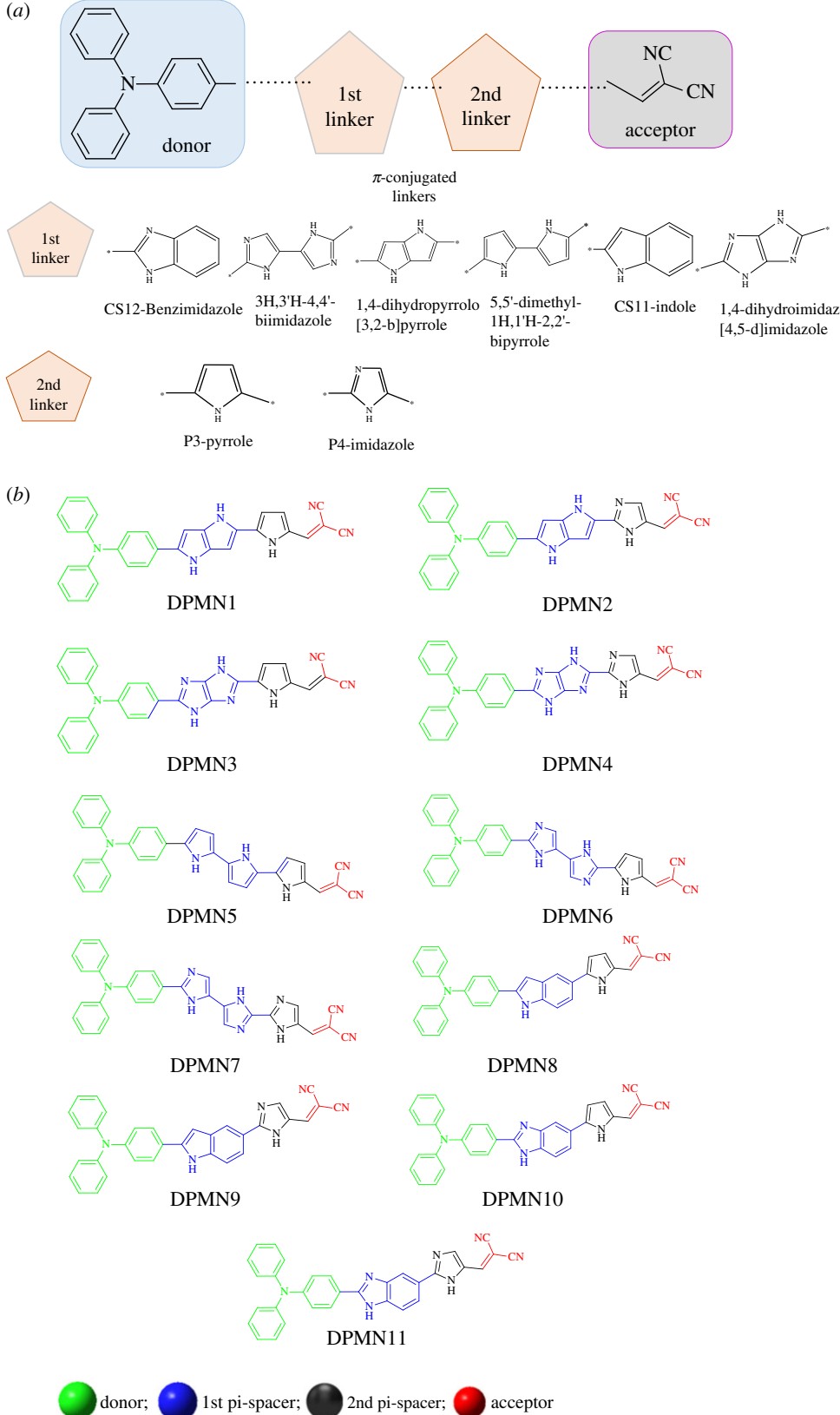

**Figure 1.** Donor, acceptor and π-conjugated linkers (*a*). Structures of the studied dyes (**DPMN1–DPMN11**) (*b*).

regarded as more reactive, less stable and soft molecules that are far more polarized and serve as a finer competitor in offering the best NLO response [35–38]. Keeping in view all these implications, the band gap between orbitals of **DPMN1–DPMN11** chromophores is determined and results are tabulated in table 1.

**Table 1.** Energies of molecular orbitals of investigated chromophores.

| dye | $E_{HOMO}$ | $E_{LUMO}$ | band gap |
|---|---|---|---|
| DPMN1 | −5.15 | −2.74 | 2.40 |
| DPMN2 | −5.24 | −2.91 | 2.32 |
| DPMN3 | −5.37 | −2.91 | 2.45 |
| DPMN4 | −5.45 | −3.09 | 2.35 |
| DPMN5 | −5.22 | −2.99 | 2.22 |
| DPMN6 | −5.43 | −2.98 | 2.45 |
| DPMN7 | −5.44 | −3.19 | 2.26 |
| DPMN8 | −5.47 | −2.76 | 2.71 |
| DPMN9 | −5.51 | −2.97 | 2.54 |
| DPMN10 | −5.59 | −2.83 | 2.77 |
| DPMN11 | −5.65 | −3.04 | 2.60 |

Data from table 1 reveal that among the pyrrole family, **DPMN1** with 1,4-dihydropyrrolo[3,2-b]pyrrole as the first π-conjugated linker expressed the least band gap (2.407 eV). Other dyes, **DPMN3**, **DPMN6**, **DPMN8** and **DPMN10** with 1,4-dihydroimidazo[4,5-d]imidazole, 3H,3'H-4,4'-biimidazole, CS11-indole and CS12-Benzimidazole pi-spacers, respectively, show a remarkably large energy gap, as shown in table 1. Overall, the increasing bandgap order of these dyes is obtained as: **DPMN1** < **DPMN3** < **DPMN6** < **DPMN8** < **DPMN10**, which revealed that 1,4-dihydropyrrolo[3,2-b]pyrrole gives the best outcomes with pyrrole spacer and reduced the energy gap.

Similarly, among **DPMN2**, **DPMN4**, **DPMN5**, **DPMN7**, **DPMN9** and **DPMN11** chromophores containing imidazole as the second π-linker, **DPMN11** exhibits the highest energy gap of 2.6 eV, and then this value starts to diminish in other compounds as π-linkers changes and the least band gap is examined in **DPMN5**. The decreasing energy gap order of the following series of dyes having imidazole is: **DPMN11** > **DPMN9** >, **DPMN4** > **DPMN2** > **DPMN7** > **DPMN5** (table 1). Interestingly, it is also examined that the dyes with imidazole pi-spacer exhibited a smaller band gap than pyrrole. This might be due to the fact that imidazole is 100 time more basic than pyrrole due to the presence of two nitrogen atoms which may enhance the resonance stabilization in the imidazole ring which in results stabilized the molecule by lowering their band gap. Overall, the highest energy gap observed is 2.767 eV in **DPMN10**, while in **DPMN5**, the lowest band gap 2.224 eV is obtained. The increasing order of energy gap of all studied chromophores is examined as: **DPMN5** < **DPMN7** < **DPMN2** < **DPMN4** < **DPMN1** < **DPMN3**, **DPMN6** < **DPMN9** < **DPMN11** < **DPMN8** < **DPMN10**. Additionally, the charge densities on the surface of orbitals are also investigated and pictographs are displayed in figure 2. For HOMO, the charge densities are concentrated at the entire molecule, while for LUMO, it is located at DCV (accepter moiety) maximally while partially over the spaces (figure 2).

## 2.3. Global reactivity parameters

The $E_{LUMO}$–$E_{HOMO}$ is used to illustrate reactivity and stability by assessing the GRP [36] such as ionization potential (IP), electron affinity (EA), electronegativity ($X$), global hardness ($\eta$), chemical potential ($\mu$), global electrophilicity ($\omega$) and global softness ($\sigma$) [36,37,39–41]. The outcomes for studied chromophores are calculated and are tabulated in electronic supplementary material, table S13. The following equations [42] are used to calculate these descriptors

$$IP = -E_{HOMO} \tag{2.1}$$

and

$$EA = -E_{LUMO} \tag{2.2}$$

where IP is the ionization potential (eV) and EA the electron affinity (eV).

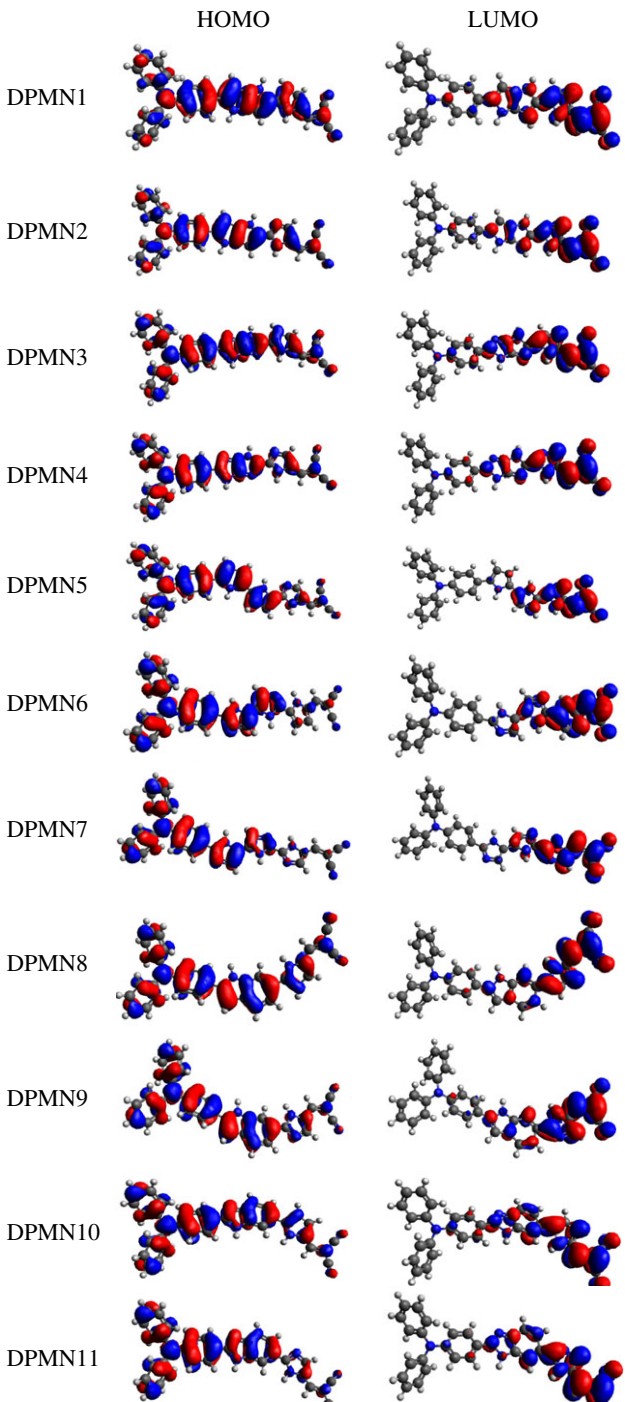

|  | HOMO | LUMO |
| --- | --- | --- |

DPMN1

DPMN2

DPMN3

DPMN4

DPMN5

DPMN6

DPMN7

DPMN8

DPMN9

DPMN10

DPMN11

**Figure 2.** HOMOs and LUMOs orbitals of the studied dyes.

Koopmans's theorem [43] was used for the calculation of the electronegativity ($X$), the chemical hardness ($\eta$) and the chemical potential ($\mu$) as

$$X = \frac{[IP + EA]}{2} = -\frac{[E_{LUMO} + E_{HOMO}]}{2},\tag{2.3}$$

$$\eta = \frac{[IP - EA]}{2} = -\frac{[E_{LUMO} - E_{HOMO}]}{2}\tag{2.4}$$

and

$$\mu = \frac{E_{HOMO} + E_{LUMO}}{2}.\tag{2.5}$$

The following equation was used for global softness ($\sigma$)

$$\sigma = \frac{1}{2\eta}. \tag{2.6}$$

The calculation of electrophilicity index ($\omega$) was reported by Parr *et al.* [36] as

$$\omega = \frac{\mu^2}{2\eta}. \tag{2.7}$$

The electron-donating and electron-accepting capabilities of compounds were characterized by ionization potential and electron affinity amplitudes, respectively. Among entitled compounds, **DPMN11** has the maximum IP value; 5.644 eV, while **DPMN1** has the minimum value; 5.151 eV. Moreover, the **DPMN7** dye has EA value; 3.19 eV and **DPMN1** has the least EA value; 2.744 eV. The IP value shows the energy required to remove an electron from a molecule. Higher IP values show more stability and chemical inertness [44]. The IP values of **DPMN1–DPMN11** were found to be higher in magnitude than EA values indicating that the designed compounds contained excellent electron-accepting ability. In order to understand the stability of molecules, chemical potential values ($\mu$) are considered [45]. The $\mu$ relates to molecular electronegativity, where its negative values show to accept electron easily [37] and $\chi$ explain the electron attraction. $\eta$ and $\sigma$ tell us about the behaviour of compounds under study in terms of energy gap [46]. In our studied chromophores, the negative values of chemical potential reflect the stability of molecules that is evident by the greater hardness values. The increasing order of global hardness is represented as: **DPMN5 < DPMN7 < DPMN2 < DPMN4 < DPMN1 < DPMN3–DPMN6 < DPMN9 < DPMN11 < DPMN8 < DPMN10**, which is the same as that of the increasing energy gap order of entitled dyes. This order is in fine agreement with the HOMO–LUMO energy gap proving the fact that molecules with large $\Delta E$ value are considered as hard molecules with greater kinetic stability, less reactivity and resistance to change in electronic configurations. Global softness is another factor in order to comprehend reactivity and stability of entitled dyes that is directly related to chemical potential. The increasing order of softness values given as: **DPMN10 < DPMN8 < DPMN11 < DPMN9 < DPMN3 = DPMN6 < DPMN1 < DPMN4 < DPMN2 < DPMN7 < DPMN5**, that is total reversal of the increasing energy gap order representing **DPMN10** (0.361402) with least value and less reactivity; however, **DPMN5** (0.44964) is the most reactive molecule with highest softness value among all others. Overall (electronic supplementary material, table S4), global reactivity descriptors have shown an excellent correlation with HOMO–LUMO band gap order. It is commonly accepted and well known that low-lying HOMO–LUMO gap might enhance the NLO response. This statement is valid in our studied systems which shows great promise for potential use of investigated compounds of their strong NLO response in optoelectronic applications.

## 2.4. Natural bond orbital analysis

NBO is a very significant tool for charge transfer interactions between empty and filled orbits [26,27]. It is often assumed that the transferring of charges is shifted from the electron-donating part to the accepter part in the D–$\pi$–A systems. Hence, to understand the charge transfer phenomena of our designed compounds (**DPMN1–DPMN11**), NBO analysis was accomplished and results are shown in table 2. This NBO investigation suggested favourable charge transfer values for all the donor moieties, indicating that our designed compounds had excellent potential for donation. Meanwhile, the negative values of NBO charges of all the acceptors revealed that all dyes can accept electrons efficiently. Moreover, the charge description in $\pi$-linkers suggested that they can provide a pathway and facilitated efficient electron transfer from donor to acceptor, except the **DPMN11** that showed a negative value of charge transfer.

Overall, the investigations indicated that charge is successfully migrated from donor towards acceptor by pi-spacers and a charge separation state was formed as the all donor and $\pi$-conjugated bridge show positive while all the acceptors exhibit negative values. The highest NBO values of charges for $\pi$-conjugated linkers is found in the compound **DPMN1** with more charge transfer properties while least values has been examined for the compound **DPMN11**. All the other designed dyes show good agreement with each other (table 2).

## 2.5. Nonlinear optical properties

Over a decade or more, the organic NLO-based dyes are given much more attention than their inorganic equivalents. The provoked attention is because of their higher fabrication and shorter response times

**Table 2.** Results of NBO charges for donor, π-spacer and acceptor for **DPMN1–DPMN11**.

| dyes | donor | π-linkers | acceptors |
|---|---|---|---|
| **DPMN1** | 0.0468 | 0.2696 | −0.3165 |
| **DPMN2** | 0.0549 | 0.2315 | −0.2864 |
| **DPMN3** | 0.0943 | 0.1886 | −0.2828 |
| **DPMN4** | 0.1025 | 0.1483 | −0.2507 |
| **DPMN5** | 0.0388 | 0.1523 | −0.2728 |
| **DPMN6** | 0.0961 | 0.1694 | −0.2655 |
| **DPMN7** | 0.0988 | 0.1341 | −0.2328 |
| **DPMN8** | 0.0764 | 0.2237 | −0.3001 |
| **DPMN9** | 0.1801 | 0.1863 | −0.2667 |
| **DPMN10** | 0.1237 | 0.1697 | −0.2934 |
| **DPMN11** | 0.1280 | −0.0144 | −0.1136 |

**Table 3.** Dipole polarizability data and major contributing tensors (*a.u.*) of the designed dyes (**DPMN1–DPMN11**).

| dye | $\alpha_{xx}$ | $\alpha_{yy}$ | $\alpha_{zz}$ | $\langle \alpha \rangle$ |
|---|---|---|---|---|
| **DPMN1** | 1242.45 | 431.51 | 255.93 | 653.30 |
| **DPMN2** | 1245.47 | 426.74 | 241.84 | 638.02 |
| **DPMN3** | 1209.41 | 424.18 | 246.01 | 526.53 |
| **DPMN4** | 1205.67 | 406.50 | 244.48 | 618.88 |
| **DPMN5** | 1206.82 | 447.52 | 270.92 | 641.75 |
| **DPMN6** | 1106.45 | 439.05 | 276.16 | 607.22 |
| **DPMN7** | 1103.19 | 419.48 | 275.50 | 599.39 |
| **DPMN8** | 1024.83 | 517.77 | 255.68 | 599.43 |
| **DPMN9** | 1052.26 | 502.56 | 246.97 | 600.60 |
| **DPMN10** | 1031.42 | 492.83 | 255.00 | 593.08 |
| **DPMN11** | 1048.11 | 468.83 | 256.61 | 591.18 |

majorly. For optoelectronic devices, optic memory systems, networking and signal manipulation, NLO products are extensively employed. Strong expertise in the NLO materials is crucial for the modulation of valuable complexes [45]. The linear polarizability and hyperpolarizability values of entitled compounds have been determined, and findings are tabulated in tables 3 and 4, respectively, to assess the impacts of various π-conjugates and π-spacers on NLO characteristics of **DPMN1–DPMN11**.

Table 3 reveals, among the pyrrole family, the highest value of $\langle \alpha \rangle$ is examined in **DPMN1** (653.30 a.u.) which has 1,4-dihydropyrrolo[3,2-b] as π linker, which reduces to 607.22 a.u. in **DPMN6**, having 3H,3′H-4,4′-biimidazole conjugated linker; further, it diminishes to 599.43 a.u. in **DPMN8**, containing CS11-indole π-linker, which decreases to 593.09 a.u. in **DPMN10**, comprising CS12-Benzimidazole as the conjugated π-linker which further apart to 526.54 a.u. in **DPMN3** having 1,4-dihydroimidazo[4,5-d]imidazole *as* π-conjugated linker. Overall, the increasing order of $\langle \alpha \rangle$ value is obtained as: **DPMN3 < DPMN10 < DPMN8 < DPMN6 < DPMN1**.

## 2.6. DPMN1–DPMN11

Similarly, the highest value of $\langle \alpha \rangle$ among the chromophores having imidazole as the second pi-spacer is observed in **DPMN5** (641.76 a.u.) having 5,5′-dimethyl-1H,1′H-2,2′-bipyrrole as the conjugated pi-linker, that lessens to 638.02 a.u. in **DPMN2** in which 1,4-dihydropyrrolo[3,2-b]pyrrole is used as the first π-linker. Further, this $\langle \alpha \rangle$ value reduces to 618.89, 600.60, 599.39 and 591.19 a.u. in **DPMN4**, **DPMN9**,

**Table 4.** The computed second-order polarizabilities ($\beta_{tot}$) and major contributing tensors (a.u) of **DPMN1–DPMN11**.

| dye | $\beta_{xxx}$ | $\beta_{xxy}$ | $\beta_{xyy}$ | $\beta_{xzz}$ | $\beta_{yzz}$ | $\beta_{zzz}$ | $\beta_{tot}$ |
|---|---|---|---|---|---|---|---|
| **DPMN1** | −52 013.951 | 3284.38 | 823.96 | 27.742 | 7.209 | −13.493 | 51 313.8 |
| **DPMN2** | 64 581.613 | −2945.596 | −1038.401 | −136.688 | −1.96 | −7.159 | 63 489.225 |
| **DPMN3** | −56 540.48 | 2903.755 | 554.336 | 75.867 | −7.722 | −19.697 | 56 015.804 |
| **DPMN4** | 70 500.324 | −2538.276 | −867.096 | −95.277 | −9.748 | 4.886 | 69 603.093 |
| **DPMN5** | 65 805.331 | 3508.612 | −778.622 | −21.824 | 6.745 | −25.2004 | 65 163.773 |
| **DPMN6** | −41 250.326 | 1604.353 | 738.158 | −198.575 | −35.436 | 24.174 | 40 847.257 |
| **DPMN7** | −53 477.565 | 2344.521 | 647.064 | 101.431 | 7.115 | 1.451 | 52 806.450 |
| **DPMN8** | −29 903.446 | 2733.562 | −1284.971 | 66.061 | −36.901 | −0.8076 | 314 412.661 |
| **DPMN9** | −43 099.362 | 3880.422 | −1268.132 | 92.366 | −34.83 | −15.713 | 44 594.584 |
| **DPMN10** | 28 927.61 | 1699.75 | 836.19 | −62.278 | −37.899 | 1.327 | 29 837.233 |
| **DPMN11** | −41 142.33 | 2444.233 | −691.129 | 13.914 | 29.707 | 5.687 | 41 979.682 |

**DPMN7** and **DPMN11** in which 1,4-dihydroimidazo[4,5-d]imidazole, CS11-indole, 3H,3′H-4,4′-biimidazole and CS12-Benzimidazole are used as the first π-linkers. The decreasing order of $\langle\alpha\rangle$ value in dyes is observed as: **DPMN5 < DPMN2 < DPMN4 < DPMN9 < DPMN7 < DPMN11**.

The HOMO–LUMO band gap inversely related with polarizability, greater polarization refers to the low energy gap. Also, it is evident that a smaller band gap and larger linear polarizability designates greater hyperpolarizability, hence referring significant NLO response [33]. Equation (2.8) illustrates the calculation of dipole polarizability

$$\alpha \propto \frac{(\mathbf{M_X^{gm}})^2}{E_{gm}}, \tag{2.8}$$

$\mathbf{M_X^{gm}}$ represents the transition moment from ground → $n_{th}$ excited state, and $E_{gm}$ refers to transition energy. A direct relationship of polarization is seen with transition moment square while inversely related with the transition energy. The transition dipole moment demonstrates the electronic transition effects and interactions of dyes with certain electromagnetic waves of the given polarization. Normally, we can say that a molecule with a higher $\mathbf{M_X^{gm}}$ value and minor $E_{gm}$ value will present a greater hyperpolarizability value. The NLO response of the first hyperpolarizability is related to intermolecular charge transfer (ICT). An excellent charger transfer is examined from donor to acceptor via π-linkers. NLO response of **DPMN1–DPMN11** with reference to second-order polarizability ($\beta$) values for **DPMN1–DPMN11** are investigated and results are displayed in table 4.

The difference in the values of $\beta_{tot}$ arises due to different kinds of π-linkers and π-spacer used. Among compounds having pyrrole as π-spacer, **DPMN8** with CS11-indole as the first π-linker expressed the highest value of $\beta_{tot}$ 314 412.66 a.u. This value decreased to 56 015.81 a.u. in **DPMN3** and became narrow to 51 313.8 a.u. in **DPMN1** as the first pi-spacer changes from 1,4-dihydroimidazo[4,5-d]imidazole to 1,4-dihydropyrrolo[3,2-b]pyrrole, respectively. This value further decreases to 40 847.26 a.u. and then to 29 837.23 a.u. in **DPMN6–DPMN10** as the first π-conjugated spacer changes from 3H,3′H-4,4′-biimidazole to CS12-Benzimidazole, respectively. The decreasing order for the $\beta_{tot}$ values of these dyes is: **DPMN8 < DPMN3 < DPMN1 < DPMN6 < DPMN10**.

Similarly, among the imidazole family, the **DPMN4** has the highest $\beta_{tot}$ value of 69 603.093 a.u. having 1,4-dihydroimidazo[4,5-d]imidazole as a π-linker. This value starts to diminish as the pi-linker changes, so the lowest value examined in **DPMN11** is 41 979.682 a.u. The decreasing order of $\beta_{tot}$ is described as **DPMN11 < DPMN9 < DPMN7 < DPMN2 < DPMN5 < DPMN4**. Consequently, it was investigated that the chromophores with imidazole second spacer expressed significantly larger NLO response than pyrrole. This might be due to the greater basic nature of imidazole as explained above. Interestingly, it was also seen that the dye **DPMN8** with pyrrole spacer exhibited larger $\beta_{tot}$ response. The first spacer CS11-indole due to its fused ring structure when combined with imidazole may interrupt the electronic transition in **DPMN9**, so it shows a lower value than **DPMN8**.

Overall, a maximum value of $\beta_{tot}$, 314 412.661(a.u.) is observed for **DPMN8**, while the smallest value is observed for **DPMN10** with 29 837.233 (a.u.). Overall, the increasing order for $\beta_{tot}$ of all dyes is:

**Table 5.** Computed transition maximum absorption wavelengths ($\lambda_{max}$), energy ($E_{ge}$/eV), oscillator strengths ($f_{os}$), transition moment ($\Delta\mu_{gm}$), LHE and transition natures of entitled compounds. MO, molecular orbital; H, HOMO; L, LUMO.

| dye | $E_{ge}$ | $\lambda_{max}$ | $f_{os}$ | LHE | $\mu_{gm}$ | major MO transitions |
|---|---|---|---|---|---|---|
| DPMN1 | 2.49 | 497 | 2.01 | 0.9901 | 5.23 | H-1 → L (13%), H → L (79%) |
| DPMN2 | 2.54 | 488 | 1.93 | 0.989 | 5.93 | H-1 → L (15%), H → L + 1 (67%) |
| DPMN3 | 2.62 | 473 | 2.15 | 0.993 | 4.97 | H-1 → L (20%), H → L (71%) |
| DPMN4 | 2.69 | 460 | 2.11 | 0.992 | 5.85 | H-1 → L (23%), H → L (64%) |
| DPMN5 | 2.66 | 466 | 1.68 | 0.980 | 6.46 | H-1 → L (23%), H → L (61%) |
| DPMN6 | 2.87 | 432 | 1.84 | 0.985 | 4.85 | H-1 → L (45%), H → L (41%) |
| DPMN7 | 2.95 | 420 | 1.74 | 0.982 | 6.11 | H-1 → L (41%), H → L (39%) |
| DPMN8 | 2.88 | 431 | 1.79 | 0.984 | 3.93 | H-1 → L (37%), H → L (51%) |
| DPMN9 | 2.89 | 429 | 1.78 | 0.983 | 5.04 | H-1 → L (34%), H → L (51%) |
| DPMN10 | 2.95 | 420 | 1.91 | 0.988 | 3.07 | H-1 → L (48%), H → L (42%) |
| DPMN11 | 2.99 | 414 | 1.93 | 0.988 | 3.97 | H-1 → L (47%), H → L (39%) |

DPMN10 < DPMN6 < DPMN11 < DPMN9 < DPMN1 < DPMN7 < DPMN3 < DPMN2 < DPMN5 < DPMN4 < DPMN8.

The $\beta_{tot}$ values of studied molecules **DPMN1–DPMN11** are compared with similar TPV-DCV-based reported molecules [17]. The highest $\beta_{tot}$ value 314 412.661(a.u.) noted in studied molecule **DPMN8** is marked as 175 337 (a.u) higher when compared with the thiazole-based reported molecule $\beta_{tot}$ value 139 075 (a.u.) [17]. The results of the remaining pyrole and imidazole-based investigated compounds **(DPMN1–DPMN11)** are also found to be in good agreement with reported thiophene and thiazole-based TPV-DCV compounds. These results provide evidence that investigated compounds, especially **DPMN8** have the potential of being used as the NLO candidate.

For further attestation of the values, $\beta_{tot}$ values are also compared with urea ($\beta_{tot}$ = 43 a.u.), considered as an organic reference molecule [46]. It is examined that $\beta_{tot}$ value of **DPMN1**, **DPMN2**, **DPMN3**, **DPMN4**, **DPMN5**, **DPMN6**, **DPMN7**, **DPMN8**, **DPMN9**, **DPMN10** and **DPMN11** are 11 193.3, 1476.49, 1302.69, 1618.68, 1515.44, 949.93, 1228.06, 7311.92, 1037.08, 693.89 and 976.27 times higher than urea, respectively.

All chromophores exhibited excellent hyperpolarizability values which indicate that structural modification by using efficient π-conjugated linkers between D and A units is a very effective technique to obtain an appealing NLO response.

## 2.7. UV–visible spectral analysis

TD-DFT computations are used for the investigation of UV–visible spectrum at CAM-B3LYP with 6-311+ G (d, p) basis set. During TD-DFT computations, the lowest singlet-singlet six energy transitions were examined (see electronic supplementary material, tables S14–S24). Calculated transition energy ($E_{ge}$), oscillator strength ($f_{os}$), nature of transitions and maximum absorption wavelength ($\lambda_{max}$) are given in table 5, while the spectra of **DPMN1–DPMN11** are displayed in figure 3. Overall, dye compounds showed the absorbance range in the UV–visible region.

In all the compounds having pyrrole as the second pi-spacer, the highest value of $\lambda_{max}$ is examined in **DPMN1** (497.67 nm). This value of maximum absorption further decreases as the pi-linker changes and the least value of maximum absorption is analysed in **DPMN10** as 420.27 nm. The decreasing order is found as **DPMN1 > DPMN3 > DPMN6 > DPMN8 > DPMN10**. The **DPMN1** dye with 4-dihydropyrrolo[3,2-b] pyrrole as the first pi-conjugated linker shows the maximum red shift among all the pyrrole family. Similarly, for the dyes having imidazole as the second pi-spacer, the minimum value of $\lambda_{max}$ investigated in **DPMN11** is 414.28 nm. This value increases as the pi-linker changes, so the largest value of $\lambda_{max}$ is studied in **DPMN2**, having 488.80 nm $\lambda_{max}$, which indicates that **DPMN2** is more bathochromic.

The computed absorption values of pyrrole and imidazole-based investigated compounds (**DPMN1–DPMN11**) are found to be in the range of 414–497 nm which is in good agreement with the reported experimental 424–497 nm range of thiophene and thiazole-based TPV-DCV molecules [17]. The

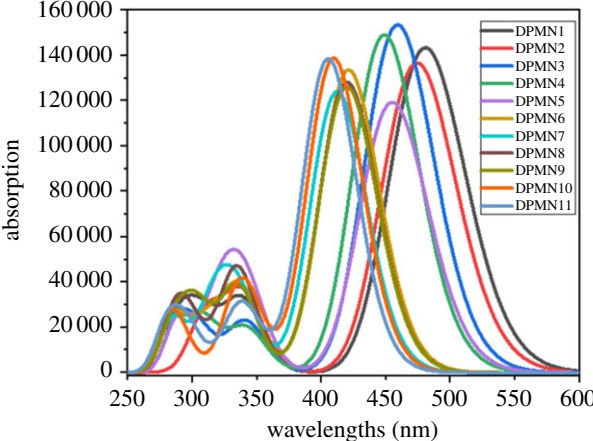

**Figure 3.** Absorption spectra of dyes (**DPMN1–DPMN11**).

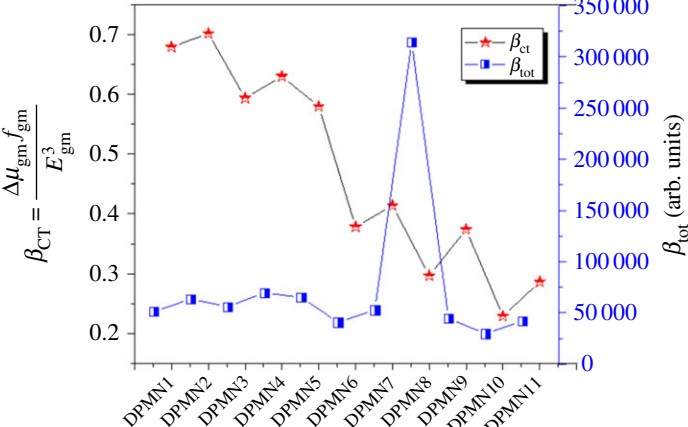

**Figure 4.** Association between the $\beta_{tot}$ (blue line) values and the corresponding $\beta_{CT}$ (red line) values for **DPMN1–DPMN11**.

synergy between computed and reported values suggests that the investigated molecules **DPMN1–DPMN11** may produce significant NLO results.

It is evident that most of the transitions (HOMO → LUMO) of electrons mainly originate from the donor (TPA) to the acceptor (DCV), including **DPMN2**, **DPMN6**, **DPMN7**, **DPMN10** and **DPMN11** chromophores. In **DPMN2**, the HOMO → LUMO + 1 transition is different from the entire series of these dyes in **DPMN6**, **DPMN7**, **DPMN10** and **DPMN11**, the transitions occur from HOMO-1 to LUMO. Another important factor affecting the optical efficiency of chromophores is LHE. The maximum photocurrent response is displayed by those compounds which have a large LHE value. Equation (2.9) for LHE calculation [47] of compounds is given below

$$LHE = 1 - 10^{-f}. \tag{2.9}$$

The above equation represents '*f*' as the oscillator strength of compounds. The LHE values for the dyes **DPMN1–DPMN11** are displayed in table 5. The highest value of LHE in the following series is 0.993 for the dye **DPMN3**, and it is the largest among all dyes. Oudar & Chemla [45] formulated a two-state model and it is extensively used in the literature to investigate the NLO response containing the critical excited and ground state in sum-over-state expression and can be represented by equation (2.10). In this model, an interaction was formed between the charge transfer transition and second-order polarizability, the origin of push–pull architecture for modelling remarkable NLO compound

$$\boldsymbol{\beta}_{CT} = \frac{\Delta\boldsymbol{\mu}_{gm}f_{gm}}{E_{gm}^3} \tag{2.10}$$

Here, $\Delta\mu_{gm}$ defines the difference between excited and ground state, and dipole moment is directly related to the second-order polarizability. The oscillator strength from the ground state and the *n*th

excited state is expressed by $f_{\mathbf{gm}}$ directly related to the $\beta$ while $E^3_{\mathbf{gm}}$, which is the transition energy in the cube, is inversely related to the $\beta$-value [48]. Transition moment and oscillation strength are significant factors in the $\beta$-value description [45], and the NLO materials with large transition moment and oscillator strength and low energy CT have shown great $\beta$-values. The good relationship between hyperpolarizability and two level model for our compounds is shown in figure 4.

The above consequences stated that manipulating the various kinds of $\pi$-bridges gives us a crucial concept in the modelling of novel D–π–A structures, which gives remarkable NLO results that can enhance the photoelectric and optical properties.

# 3. Conclusion

Triphenylamine–dicyanovinylene-based (**DPMN1–DPMN11**) chromophores were theoretically designed by structural tailoring with various pi-spacers and the influence of these spacers on NLO properties was examined. The findings reveal that pi-linker expressed promising effect over the D–P–A architecture which strongly tuned electronic, phtophysical and NLO properties of designed chromophores. All investigated compounds exhibited the broader absorption spectrum (in the range of 414–497 nm) and larger LHE with least transitional energy (2.49–2.99 eV). Highest red shift ($\lambda_{\max} = 497.67$ nm) was investigated in **DPMN1**. The FMO studies revealed that HOMO is migrated over TPA and partially on $\pi$-linkers or acceptors. By contrast, most LUMOs are mounted on DCV (acceptor) and partly on $\pi$-conjugates. Moreover, the least band gap 2.22–2.60 eV was studied in **DPMN1–DPMN11**, respectively. Additionally, NBO findings elucidated that electrons are efficiently transferred through $\pi$-linker from TPA to DCV, leading to the establishment of a charge transferring state. Examination of electronic transitions from donor to acceptor moieties via $\pi$-conjugated linkers revealed greater linear ($\langle\alpha\rangle$ = 526.536–641.756 a.u.) and nonlinear ($\beta_{\mathrm{tot}}$ = 51 313.8–314 412.661 a.u.) response. Interestingly, compounds with imidazole spacer expressed lower band gap and higher NLO properties due to its higher basic nature when compared with pyrrole. Overall, all the designed dyes have shown conspicuous NLO response with higher polarizability and first hyperpolarizability values especially **DPMN8** exhibit highest $\beta_{\mathrm{tot}}$ value 314 412.661 (a.u.) which is 175 337 (a.u.) higher when compared with the thiazole-based reported molecule $\beta_{\mathrm{tot}}$ value 139 075 (a.u.). It is also examined that the $\beta_{\mathrm{tot}}$ value of **DPMN1–DPMN11** are found to be 693.89–7311.92 times higher than standard urea. These organic metal-free dyes based on this D–π–A framework are crucial in the area of research and provide new insight into experiments for the production of high-performance NLO materials.

Data accessibility. Data associated with the review article are available from: http://dx.doi.org/10.5061/dryad.4tmpg4f99 [49]. The data are provided in electronic supplementary material [50].

Authors' contributions. M.K.: conceptualization and project administration; I.S.: formal analysis; A.A.C.B. and M.I.: funding acquisition; M.U.K. and A.A.: methodology; R.H. and M.f.u.R.: software; M.f.u.R., M.S.A. and M.K.: supervision, writing—review and editing.

Competing interests. There are no conflicts to declare.

Funding. M.I. is thankful to the Dean of Scientific Research at King Khalid University Saudi Arabia for funding this research through grant no. R.G.P. 2/28/42. A.A.C.B. (grant nos. 2015/01491-3 and 2014/25770-6) is thankful to Fundacão de Amparo à Pesquisa do Estado de Sao Paulo for financial support. A.A.C.B. (grant no. 312550/2020-0) also thanks the Brazilian National Research Council (CNPq) for financial support. This study was financed in part by the Coordenação de Aperfeiçoamento de Pessoal de Nivel Superior do Brasil (CAPES) Finance Code 001.

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
