## [Peer Review File · Royal Society Open Science]

Review History

RSOS-210570.R0 (Original submission)

Review form: Reviewer 1

Is the manuscript scientifically sound in its present form?

No

Are the interpretations and conclusions justified by the results?

No

Is the language acceptable?

Yes

Do you have any ethical concerns with this paper?

No

Have you any concerns about statistical analyses in this paper?

No

Recommendation?

Major revision is needed (please make suggestions in comments)

Comments to the Author(s)

The manuscript [Structural Modulation of π -Conjugated Linkers in D- π -A Dyes Based on Triphenylamine Dicyanovinylene Framework to Explore the NLO Properties] reported a study of Triphenylamine-dicyanovinylene based chromophores by DFT calculation. Studied content include ground and Electronic Structure, Ionization potential and Electron affinity chemical reactivity parameters, NBO and NLO and so on. The research topics and contents belong to the fields of chemistry, physics and materials. The results provide an understanding the photophysical and photochemical properties of chromophores. The reviewer has the following comments and questions for the authors to address:

1. The reasons why the authors choose these chromophores (DPMN1-DPMN11) need to be explained clearly in the Introduction. Please add this introduction.
2. In methods, authors should cite the corresponding reference for equation.
3. We know that molecular structure has a great influence on ICT, so we hope that the author will discuss dihedral angle in the section of structure discussion. At the same time, the author's explanation of ICT can be referred to article " Enhancement of one- and two-photon absorption and visualization of intramolecular charge transfer of pyrenyl-contained derivatives".
4. The author thus said "Typically, HOMO is regarded as an electron donator, While LUMO is generally considered as an electron acceptor" it's not right to describe HOMO and LUMO, because we know that electron transfer is more than just an interaction between these two energy levels.
5. This rough expression of IP and EA is worth discussing, please cite the article to prove. At the same time, the author needs to specify the physical and chemical meaning of those parameters [such as electronegativity, global Hardness, chemical potential, global Electrophilicity (ω) and Global Softness] for studying NLO and other corresponding properties.
6. A more detail discussion on the result should be presented; especially, simulated absorption should be compared with experimental results. Author should add content of comparative analysis.
7. According to the data provided by the author, there is no obvious NLO properties for this series of dyes. It is suggested for authors to compare those result with similar molecules NLO to confirm the advantages of the designed molecular properties
8. The conclusion part of the manuscript does not show the highlights of this work. Please rewrite it.

Review form: Reviewer 2 (Zahid Shafiq)**Is the manuscript scientifically sound in its present form?**

Yes

Are the interpretations and conclusions justified by the results?

Yes

Is the language acceptable?

Yes

Do you have any ethical concerns with this paper?

No

Have you any concerns about statistical analyses in this paper?

No

Recommendation?

Accept with minor revision (please list in comments)

Comments to the Author(s)

The authors have reported manuscript entitled "Structural Modulation of π -Conjugated Linkers in D- π -A Dyes Based on Triphenylamine

Dicyanovinylene Framework to Explore the NLO Properties" In this work, donor- π -acceptor type series of Triphenylamine-dicyanovinylene based chromophores (DPMN1- DPMN11) was designed theoretically by the structural tailoring of π -linkers to exploit changes in the optical properties and their NLO behavior. Frontier molecular orbitals (FMOs), Global reactivity parameters (GRP), NBO (natural bond orbital) analysis, absorption spectra and molecular nonlinear optical (NLO) have been analyzed and reported. Computational chemistry has been used by Khalid and coworkers to evaluate structures of different compounds by using principles of theoretical and quantum chemistry integrated into useful computer programs.

The development of nonlinear optical (NLO) compounds is an important field of research in recent years. Good scientific effort has been made by theoretical scientists. The research article is properly handled by the authors. To sum up, I believe that the idea and the results presented in the paper are interesting and deserve publication in Royal Society Open Science and can be of interest for its readership. However, I recommend following minor points to consider before final publication:

Comments:

- Abstract: by the structural tailoring of π -linkers of..... Write down the experimental compound name here.
- Optical spellings are incorrect.
- Line 39. Please cite reference after "fiber optics, telecommunication and information technology"
- The structures and Compound names in Figure A are not visible. It should be reproduced for better readership. The font size and style should be same for better view and also the colours must be bright for better visibility.
- At some places for example (Page 3, Line 47), (Page 4, Line 7, 10) word [R] is mentioned. Authors should correct it.
- The compounds name is bold at some position while isn't bold-faced at some places (Page 6). Authors should maintain the similarity.
- Equation number is missing for light harvesting efficiency equation.
- Authors should correct equation number 11.
- Please indicate the full name of each abbreviation at its first appearance in the draft.
- The authors should use all symbols in italic format throughout the manuscript.
- The size of the figures in the manuscript is needed to properly arrange to make it suitable for reading.
- All references should be checked again. Please verify the consistency in the style of the references list. The abbreviations used in references should be rechecked.

Decision letter (RSOS-210570.R0)

Dear Dr Akram:

Title: Structural Modulation of π -Conjugated Linkers in D- π -A Dyes Based on Triphenylamine Dicyanovinylene Framework to Explore the NLO Properties
Manuscript ID: RSOS-210570

The editor assigned to your manuscript has now received comments from reviewers. We would like you to revise your paper in accordance with the referee and Subject Editor suggestions which can be found below (not including confidential reports to the Editor). Please note this decision does not guarantee eventual acceptance.

Please submit your revised paper before 04-Jun-2021. Please note that the revision deadline will expire at 00.00am on this date. If we do not hear from you within this time then it will be assumed that the paper has been withdrawn. In exceptional circumstances, extensions may be possible if agreed with the Editorial Office in advance. We do not allow multiple rounds of revision so we urge you to make every effort to fully address all of the comments at this stage. If deemed necessary by the Editors, your manuscript will be sent back to one or more of the original reviewers for assessment. If the original reviewers are not available we may invite new reviewers.

Royal Society of Chemistry
Thomas Graham House
Science Park, Milton Road
Cambridge, CB4 0WF

Royal Society Open Science - Chemistry Editorial Office

On behalf of the Subject Editor Professor Anthony Stace and the Associate Editor Professor Chaohua Cui.

RSC Associate Editor:
Comments to the Author:
(There are no comments.)

RSC Subject Editor:
Comments to the Author:
(There are no comments.)

Reviewers' Comments to Author:

Reviewer: 1

Comments to the Author(s)

The manuscript [Structural Modulation of π -Conjugated Linkers in D- π -A Dyes Based on Triphenylamine Dicyanovinylene Framework to Explore the NLO Properties] reported a study of Triphenylamine-dicyanovinylene based chromophores by DFT calculation. Studied content include ground and Electronic Structure, Ionization potential and Electron affinity chemical reactivity parameters, NBO and NLO and so on. The research topics and contents belong to the fields of chemistry, physics and materials. The results provide an understanding the photophysical and photochemical properties of chromophores. The reviewer has the following comments and questions for the authors to address:

1. The reasons why the authors choose these chromophores (DPMN1-DPMN11) need to be explained clearly in the Introduction. Please add this introduction.
2. In methods, authors should cite the corresponding reference for equation.
3. We know that molecular structure has a great influence on ICT, so we hope that the author will discuss dihedral angle in the section of structure discussion. At the same time, the author's explanation of ICT can be referred to article "Enhancement of one- and two-photon absorption and visualization of intramolecular charge transfer of pyrenyl-contained derivatives".
4. The author thus said "Typically, HOMO is regarded as an electron donator, While LUMO is generally considered as an electron acceptor" it's not right to describe HOMO and LUMO, because we know that electron transfer is more than just an interaction between these two energy levels.
5. This rough expression of IP and EA is worth discussing, please cite the article to prove. At the same time, the author needs to specify the physical and chemical meaning of those parameters [such as electronegativity, global Hardness, chemical potential, global Electrophilicity (ω) and Global Softness] for studying NLO and other corresponding properties.
6. A more detail discussion on the result should be presented; especially, simulated absorption should be compared with experimental results. Author should add content of comparative analysis.
7. According to the data provided by the author, there is no obvious NLO properties for this series of dyes. It is suggested for authors to compare those result with similar molecules NLO to confirm the advantages of the designed molecular properties
8. The conclusion part of the manuscript does not show the highlights of this work. Please rewrite it.

Reviewer: 2

Comments to the Author(s)

The authors have reported manuscript entitled “Structural Modulation of π -Conjugated Linkers in D- π -A Dyes Based on Triphenylamine

Dicyanovinylene Framework to Explore the NLO Properties” In this work, donor- π -acceptor type series of Triphenylamine-dicyanovinylene based chromophores (DPMN1- DPMN11) was designed theoretically by the structural tailoring of π -linkers to exploit changes in the optical properties and their NLO behavior. Frontier molecular orbitals (FMOs), Global reactivity parameters (GRP), NBO (natural bond orbital) analysis, absorption spectra and molecular nonlinear optical (NLO) have been analyzed and reported. Computational chemistry has been used by Khalid and coworkers to evaluate structures of different compounds by using principles of theoretical and quantum chemistry integrated into useful computer programs.

The development of nonlinear optical (NLO) compounds is an important field of research in recent years. Good scientific effort has been made by theoretical scientists. The research article is properly handled by the authors. To sum up, I believe that the idea and the results presented in the paper are interesting and deserve publication in Royal Society Open Science and can be of interest for its readership. However, I recommend following minor points to consider before final publication:

Comments:

- Abstract: by the structural tailoring of π -linkers of..... Write down the experimental compound name here.
- Optical spellings are incorrect.
- Line 39. Please cite reference after “fiber optics, telecommunication and information technology”
- The structures and Compound names in Figure A are not visible. It should be reproduced for better readership. The font size and style should be same for better view and also the colours must be bright for better visibility.
- At some places for example (Page 3, Line 47), (Page 4, Line 7, 10) word [R] is mentioned. Authors should correct it.
- The compounds name is bold at some position while isn't bold-faced at some places (Page 6). Authors should maintain the similarity.
- Equation number is missing for light harvesting efficiency equation.
- Authors should correct equation number 11.
- Please indicate the full name of each abbreviation at its first appearance in the draft.
- The authors should use all symbols in italic format throughout the manuscript.
- The size of the figures in the manuscript is needed to properly arrange to make it suitable for reading.
- All references should be checked again. Please verify the consistency in the style of the references list. The abbreviations used in references should be rechecked.

Author's Response to Decision Letter for (RSOS-210570.R0)

See Appendix A.

RSOS-210570.R1 (Revision)

Review form: Reviewer 1

Is the manuscript scientifically sound in its present form?

Yes

Are the interpretations and conclusions justified by the results?

Yes

Is the language acceptable?

Yes

Do you have any ethical concerns with this paper?

No

Have you any concerns about statistical analyses in this paper?

No

Recommendation?

Accept as is

Comments to the Author(s)

The revised manuscript could be accepted.

Review form: Reviewer 2 (Zahid Shafiq)

Is the manuscript scientifically sound in its present form?

Yes

Are the interpretations and conclusions justified by the results?

Yes

Is the language acceptable?

Yes

Do you have any ethical concerns with this paper?

No

Have you any concerns about statistical analyses in this paper?

No

Recommendation?

Accept as is

Comments to the Author(s)

The authors have revised / justified the comments raised and manuscript is now suitable for publication in RSOC.

Decision letter (RSOS-210570.R1)

Dear Dr Akram:

Title: Structural Modulation of π -Conjugated Linkers in D- π -A Dyes Based on Triphenylamine Dicyanovinylene Framework to Explore the NLO Properties
Manuscript ID: RSOS-210570.R1

It is a pleasure to accept your manuscript in its current form for publication in Royal Society Open Science. The chemistry content of Royal Society Open Science is published in collaboration with the Royal Society of Chemistry.

On behalf of the Subject Editor Professor Anthony Stace and the Associate Editor Professor Chaohua Cui.

RSC Associate Editor:
Comments to the Author:
(There are no comments.)

RSC Subject Editor:
Comments to the Author:
(There are no comments.)

Reviewer(s)' Comments to Author:

Reviewer: 2

Comments to the Author(s)

The authors have revised / justified the comments raised and manuscript is now suitable for publication in RSOC.

Reviewer: 1

Comments to the Author(s)

The revised manuscript could be accepted.

Date: May 28, 2021

Publishing Editor,
Royal Society Open Science
Chemistry Editorial Office

Royal Society of Chemistry
Thomas Graham House
Science Park
Milton Road
Cambridge.
CB4 0WF

Dr. Muhammad Safwan Akram
Principal Lecturer/Associate Professor
Enterprise & External Engagement
National Horizons Centre

School of Health and Life Sciences
Teesside University, Middlesbrough
Tees Valley TS1 3BA, UK

T: +44 (0) 7766816070
T: +44 (0) 1642348102
E: safwan.akram@tees.ac.uk

Subject: Submission of the Revised Manuscript RSOS-210570

Dear Dr. Laura Smith

Thanks for the consideration of the manuscript titled “**Structural Modulation of π -Conjugated Linkers in D- π -A Dyes Based on Triphenylamine Dicyanovinylene Framework to Explore the NLO Properties**”.

The manuscript has been revised according to the comments of all reviewers. Point to point response against all the comments raised by the reviewers is given below. All the authors are very thankful to the reviewers’ comments, which have enabled us to raise the level of this manuscript. I hope that this revised version would be acceptable for publication.

With best regards,

Dr. Muhammad Safwan Akram, MRSC
PhD Cantab, MPhil (MIT/Cantab)

Answer to reviewers' comments to author:

Reviewer 1

We are grateful to the reviewer for taking time to provide valuable comments and suggestions. Below we describe our responses (highlighted in yellow text) point-by-point to each comment. Similarly, we have indicated revisions in the updated manuscript by a yellow highlight.

Comments to the Author:

The manuscript [Structural Modulation of π -Conjugated Linkers in D- π -A Dyes Based on Triphenylamine Dicyanovinylene Framework to Explore the NLO Properties] reported a study of Triphenylamine-dicyanovinylene based chromophores by DFT calculation. Studied content include ground and Electronic Structure, Ionization potential and Electron affinity chemical reactivity parameters, NBO and NLO and so on. The research topics and contents belong to the fields of chemistry, physics and materials. The results provide an understanding the photophysical and photochemical properties of chromophores. The reviewer has the following comments and questions for the authors to address:

1. The reasons why the authors choose these chromophores (DPMN1-DPMN11) need to be explained clearly in the Introduction. Please add this introduction.

Answer: We are deeply grateful to respected reviewer for the nice suggestion. The reason of the selection of DPMN1-DPMN11 compounds and the configurations has been improved in the revised draft. It is reproduced here:

We have previously published improved π bridges through double heteroaromatic rings and demonstrated their use in Triphenylamine-Dicyanovinylene dyes¹⁷. This **manuscript demonstrate** that π bridge modification is a convenient strategy to augment NLO response and to design novel NLO materials. The use of charge transfer (CT), between the electron donor and the withdrawal group, to build a new donor- π -acceptor system, **is capable of minimizing** the bandgap and regulate the transitions utilising various donor or acceptor moieties with large first hyperpolarizability values (β_{total})^{11,13-15,20-22}. Inspired from these reported strategies, the electronic properties of D- π -A system with the new pi-conjugated system is introduced in the system consisting of 1H,1'H-2,2'-bipyrrole, 1,4 dihydropyrrolo[3,2-b]pyrrole, indole, 1,4-dihydroimidazo[4,5-d]imidazole, 3H,3'H-4,4'-biimidazole, and benzimidazole referred to as initial π -spacer, and two conjugates, pyrrole and imidazole as second π -linker between the Donor, Triphenylamine (TPA) and acceptor, Dicyanovinylene (DCV). TPA, a donor unit due to its capacity for electron donation and charge transfer, is used in many hole transport materials^{23, 24}. Eleven new D- π -A type, TPA-DCV dyes DPMN1-DPMN11, have been developed with various **configuration** of first and second π -conjugates.

2. In methods, authors should cite the corresponding reference for equation.

Answer: Thanks for the suggestion, corresponding reference for the equation has been cited in revised manuscript. Please see citation number 26.

3. We know that molecular structure has a great influence on ICT, so we hope that the author will discuss dihedral angle in the section of structure discussion. At the same time, the author's explanation of ICT can be referred to article "Enhancement of one- and two-photon absorption and visualization of intramolecular charge transfer of pyrenyl-contained derivatives".

Answer: According to the instructions of respected reviewer, dihedral angle discussion has been made in structural discussion. Furthermore, the suggested paper is also cited in revised manuscript. Please see citation number 16.

The dihedral angle between C-C-C in benzene ring of TPA molecule is found to be 117° in all investigated molecules (DPMN1-DPMN11). The bond angle between C-C-N of 1,4-dihydropyrrolo[3,2-b]pyrrole attached to pyrrole in DPMN1 and imidazole in DPMN2 is found to be 106° and 110° respectively. In DPMN3 and DPMN4, C-N-N bond angle of 1,4-dihydroimidazo[4,5-d]imidazole attached to TPA side benzene ring is found to be similar at 112°. The dihedral angle between C-C-N (pyrrole) of DPMN3 and in C-N-N (imidazole) of DPMN4 are noted as 107° and 111° respectively. In DPMN5, the dihedral bond angle between C-C-N of 1H,1'H-2,2'-bipyrrole is observed as 106°. Slight increase in bond angle to 110° in C-N-N of imidazole is noted when 1H,1'H-2,2'-bipyrrole is attached to imidazole unit. 104° dihedral bond angle is marked in C-N-N of 3H,3'H-4,4'-biimidazole in DPMN6. The dihedral angle of 107° is observed for C-C-N of pyrrole unit in DPMN6. In compound DPMN7, 104° and 111° dihedral angle is found between C-N-N of 3H,3'H-4,4'-biimidazole and imidazole respectively. In DPMN8 and DPMN9, five membered C-C-N of indole towards TPA side exhibited 108° dihedral angle, while benzene ring C-C-C of indole towards pyrrole and imidazole exhibited 120° dihedral angle. 106° and 110° dihedral angle are found between C-C-N in DPMN10 pyrrole unit and C-N-N in DPMN11 imidazole unit respectively.

4. The author thus said "Typically, HOMO is regarded as an electron donor, while LUMO is generally considered as an electron acceptor" it's not right to describe HOMO and LUMO, because we know that electron transfer is more than just an interaction between these two energy levels.

Answer: Thanks to the reviewer for thorough reading. These sentences have been removed in the revised draft for clarity.

5. This rough expression of IP and EA is worth discussing, please cite the article to prove. At the same time, the author needs to specify the physical and chemical meaning of those parameters [such as electronegativity, global Hardness, chemical potential, global Electrophilicity (ω) and Global Softness] for studying NLO and other corresponding properties.

Answer: We are profoundly thankful to respected reviewer for detailed study of the manuscript. The reference for IP and EA [A. Szabo and N. S. Ostlund, Modern quantum chemistry: introduction to advanced electronic structure theory, Courier Corporation, 2012.] is cited in the revised version. The physical and chemical meanings of global reactivity parameter for studying NLO properties is mentioned in the revised draft as following lines:

"The electron-donating and electron-accepting capabilities of compounds were characterized by ionization potential and electron affinity amplitudes, respectively. The IP values of DPMN1-DPMN11 were examined higher in magnitude than EA values which indicated that our

compounds contained excellent electron-accepting ability. In order to understand the stability of molecules, chemical potential values (μ) are considered ⁴⁵. The μ relates to molecular electronegativity, where its negative values show to accept electron easily ³⁷ and χ explain the electron attraction. η and σ tell us about the behaviour of compounds under study in term of energy gap ⁴⁶. In our studied chromophores, the negative values of chemical potential reflect the stability of molecules that is evident by the greater hardness values. This order is in fine agreement with the HOMO-LUMO energy gap proving the fact that molecules with large ΔE value are considered as hard molecules with greater kinetic stability, less reactivity and resistance to change in electronic configurations. Global softness is another factor in order to comprehend the reactivity and stability of entitled dyes that is directly related to chemical potential. It is commonly accepted and well known that low lying HOMO-LUMO gap might enhance the NLO response. This statement is valid in our studied systems which is a strong clue for potential usage of the investigated compound as fine NLO candidates in optoelectronic technologies.

6. A more detail discussion on the result should be presented; especially, simulated absorption should be compared with experimental results. Author should add content of comparative analysis.

Answer: According to the instructions of respected reviewer, discussion has been added in the revised manuscript as “The computed absorption values of pyrole and imidazole based investigated compounds (DPMN1-DPMN11) are found to be in the range of 414-497 nm which is in good concurrence with the reported 424-497 nm range of thiophene and thiazole based TPV-DCV molecules ¹⁷. This harmony exists between computed and reported values suggest that the investigated molecules DPMN1-DPMN11 may produce significant NLO results”.

7. According to the data provided by the author, there is no obvious NLO properties for this series of dyes. It is suggested for authors to compare those result with similar molecules NLO to confirm the advantages of the designed molecular properties

Answer: According to the instructions of respected reviewer, NLO results have been compared with similar type of reported molecules and discussion is modified as “The β_{tot} values of studied molecules **DPMN1-DPMN11** are compared with similar TPV-DCV based reported molecules ¹⁷. The highest β_{tot} value 314412.661(a.u) noted in studied molecule **DPMN8** is marked as 175,337 (a.u) higher as compared to the thiazole based reported molecule β_{tot} value 139075 (a.u) ¹⁷. The results of remaining pyrole and imidazole based investigated compounds (**DPMN1-DPMN11**) are also found in good agreement with reported thiophene and thiazole based TPV-DCV compounds. These results provide evidence that investigated compounds, especially **DPMN8** has the potential of being used as an NLO candidate”.

8. The conclusion part of the manuscript does not show the highlights of this work. Please rewrite it.

Answer: According to the instructions of respected reviewer, conclusion part is modified with highlights of the work.

Reviewer 2

We extremely thankful to the reviewer for their comments on the manuscript and for supporting the importance of the findings reported and conclusions reached; the useful remarks and questioning are addressed below (in yellow text). In addition, we indicate revisions in the updated manuscript by a yellow highlighter in the manuscript.

Comments to the Author:

The authors have reported manuscript entitled “Structural Modulation of π -Conjugated Linkers in D- π -A Dyes Based on Triphenylamine Dicyanovinylene Framework to Explore the NLO Properties” In this work, donor- π -acceptor type series of Triphenylamine-dicyanovinylene based chromophores (DPMN1- DPMN11) was designed theoretically by the structural tailoring of π -linkers to exploit changes in the optical properties and their NLO behavior. Frontier molecular orbitals (FMOs), Global reactivity parameters (GRP), NBO (natural bond orbital) analysis, absorption spectra and molecular nonlinear optical (NLO) have been analyzed and reported. Computational chemistry has been used by Khalid and coworkers to evaluate structures of different compounds by using principles of theoretical and quantum chemistry integrated into useful computer programs.

The development of nonlinear optical (NLO) compounds is an important field of research in recent years. Good scientific effort has been made by theoretical scientists. The research article is properly handled by the authors. To sum up, I believe that the idea and the results presented in the paper are interesting and deserve publication in Royal Society Open Science and can be of interest for its readership. However, I recommend following minor points to consider before final publication:

Answer: We are profoundly thankful to the respected reviewer for the positive recommendation!

Comments:

- Abstract: by the structural tailoring of π -linkers of..... Write down the experimental compound name here.

Answer: Experimental compounds names DTTh and DTTz have been added to the abstract.

- Optical spellings are incorrect.

Answer: We are much obliged to the respected reviewer for thorough study of the manuscript. Correction has been made in the revised version.

- Line 39. Please cite reference after “fiber optics, telecommunication and information technology”

Answer: As per instructions,

1. Z. Peng and L. Yu, *Macromolecules*, 1994, **27**, 2638-2640.

2. N. Tsutsumi, M. Morishima and W. Sakai, *Macromolecules*, 1998, **31**, 7764-7769.

3. E. M. Breitung, C.-F. Shu and R. J. McMahon, *Journal of the American Chemical Society*, 2000, **122**, 1154-1160.

references are cited in the revised manuscript.

- The structures and Compound names in Figure A are not visible. It should be reproduced for better readership. The font size and style should be same for better view and also the colours must be bright for better visibility.

Answer: Sorry for inconvenience! We have gone through all the figures to improve their quality for better view and visibility. Figure A is reproduced for better readership.

- At some places for example (Page 3, Line 47), (Page 4, Line 7, 10) word [R] is mentioned. Authors should correct it.

Answer: Thanks to the respected reviewer for such detailed study of the manuscript. The word [R] was due to an error in the software. Correction has been made in the revised version.

- The compounds name is bold at some position while isn't bold-faced at some places (Page 6). Authors should maintain the similarity.

Answer: Correction is made in the revised draft. All compounds name is bold-faced now in the revised version.

- Equation number is missing for light harvesting efficiency equation.

Answer: Equation number 11 is allotted to light harvesting equation.

- Authors should correct equation number 11.

Answer: Equation number 11 is corrected as equation 12

- Please indicate the full name of each abbreviation at its first appearance in the draft.

Answer: According to the instructions of respected reviewer, the full name of each abbreviation is mentioned at its first appearance in the draft.

- The authors should use all symbols in italic format throughout the manuscript.

Answer: According to the instructions of valuable reviewer, symbols are used in italic format in the revised version.

- The size of the figures in the manuscript is needed to properly arrange to make it suitable for reading.

Answer: We have gone through all the figures to improve their quality for better view and visibility. Size of the figures has been properly arranged for suitable reading.

All references should be checked again. Please verify the consistency in the style of the references list. The abbreviations used in references should be rechecked.

Answer: We have gone through the reference section again and carefully to verify the consistency in the style and abbreviation used. All references are cited in consistent and according to journal style.

We are profoundly thankful to respected reviewer for the positive recommendation and valuable suggestions.